# Variational Set Operator Networks: Uncertainty-Aware Meta-Learning via Probabilistic Neural Operators

## Abstract

We introduce a probabilistic neural operator framework for learning conditional distributions over functions from sample observations. The proposed model, the Variational Set Operator Network (VSON), extends set-based operator learners by incorporating an amortised latent representation of the branch outputs that induces predictive distributions over function values conditioned on arbitrary sets of input–output pairs. Uncertainty is represented through a learned variational latent structure implemented with expressive normalising flows, allowing the model to capture non-Gaussian behaviour. The resulting operator produces smooth function predictions and coherent joint samples over target sets. VSON improves predictive uncertainty calibration across benchmarks while remaining competitive on accuracy, matching or exceeding the baselines on the regression and optimisation tasks.

## 1 Introduction

Operator learning methods such as Deep Operator Networks (Lu et al., 2021) and the Fourier Neural Operator (Li et al., 2021) have demonstrated strong empirical performance for learning mappings between function spaces, but they are typically defined on fixed input discretisations and therefore do not naturally accommodate variable-sized context sets. This limitation matters in applications where sensors are expensive or where measurement locations must be chosen sequentially. Such settings are common wherever each observation carries a real cost and decisions must be made from sparse, adaptively gathered data, such as Bayesian optimisation of expensive black-box functions, sequential experimental design and active learning, and surrogate modelling of physical systems (Shahriari et al., 2016). In each case a model must return not just an estimate at a new location but a trustworthy account of its own uncertainty, from a variable-sized and often small set of observations. In this paper we position neural operators within a meta-learning framework, learning a map from sparse observations of a function to itself while accounting for uncertainty. Formally, given a variable-sized context set of observed input-output pairs

$$\mathcal{C} = \{(x_i, y_i)\}_{i=1}^N, \tag{1}$$

which come from an unknown underlying function $f : \mathcal{X} \mapsto \mathcal{Y}$ itself belonging to a family of functions $\mathcal{F}$, we address how to learn a fast, generalisable conditional posterior $p(y \mid x, \mathcal{C})$. Rather than fitting a new posterior from scratch for each task, we learn a mapping from context sets to predictive posteriors using representative training instances from $\mathcal{Y}$. At test time the learned operator returns cheap posterior samples for arbitrary, variable-size context sets.

To realise this we build on permutation-invariant branch encoders and introduce the *Variational Set Operator Network* (VSON): a variational operator network whose branch is explicitly set-based. The training objective balances latent regularisation and calibration. We benchmark VSON against various Neural Process baselines on 1D regression, image completion and Bayesian optimisation (BO) tasks. VSON consistently improves uncertainty calibration, and matches or exceeds the baselines on predictive accuracy in the regression and BO settings, while trading a small amount of accuracy for substantially better calibration on image completion.

## 1.1 Related work

**Neural Processes and amortised learning** Neural Processes (NPs) and their attention and transformer extensions (ANPs, TNPs) provide a direct meta-learning approach to conditional function modelling by mapping context sets to predictive distributions (Garnelo et al., 2018b; Kim et al., 2019; Nguyen & Grover, 2023; Lee et al., 2020). This is an instance of *amortised inference*, whereby rather than solving a separate inference problem for each new task, a single network is trained to map observations directly to a predictive distribution, so that inference at test time reduces to a forward pass. Transformer neural processes (TNPs) in particular recast the conditional prediction problem as a sequence modelling task using transformer blocks and masking strategies, improving likelihood-based training and conditional fidelity over prior work. Our work adopts the same meta-learning objective but instantiates this objective in the operator-learning paradigm. There is also an overlap with the amortised sequential design literature (Huang et al., 2025; Müller et al., 2023; Li et al., 2025), particularly with respect to the BO experiment, as well as more general amortised Bayesian inference models (Müller et al., 2024; Chang et al., 2025). While these methods typically treat inference as a sequence-to-sequence task, VSON reformulates the problem as one of operator learning, enabling resolution-independent function estimation with non-Gaussian uncertainty.

**Uncertainty-aware neural operators** Uncertainty-aware neural operators have attracted attention as a natural progression of neural operators for modelling primarily physical systems. VB-DeepONet (Garg & Chakraborty, 2022) treats DeepONet weights in a variational Bayesian framework, enabling posterior uncertainty for parametric PDEs and showing improved generalisation compared to deterministic Deep-ONet. B-DeepONet (Lin et al., 2022) uses replica-exchange stochastic gradient Langevin dynamics to train a Bayesian DeepONet that handles noisy data more robustly and estimates uncertainty reliably. $\alpha$-VI Deep-ONet (Lone et al., 2024) generalises variational inference using Rényi's $\alpha$-divergence, aiming for greater robustness to prior misspecification while quantifying uncertainty in operator predictions.

Permutation-invariant function learning has also been extended to neural operator surrogates. (Prasthofer et al., 2022) introduced a variable-input version of the original DeepONet architecture which extends the branch network to contain a learned, permutation-invariant encoder. (Tretiakov et al., 2025) introduced a branch network architecture based concretely on DeepSets (Zaheer et al., 2018) which passes each context pairing through an MLP before aggregating the obtained representations via either a mean pooling or an attention mechanism. While these SetONet variants successfully handle variable-sized inputs, they remain fundamentally deterministic mappings. More recently Ma et al. (2025), proposed a conditional variational autoencoder DeepSet neural operator, which split the branch network into deterministic and small latent components, not unlike the latent variable present in the NPs, but which was limited to modelling processes with Gaussian noise. VSON builds upon the architectural backbone of SetONets but elevates it to a flexible probabilistic framework, replacing the deterministic branch output with a variational latent structure.

## 2 Background

The problem tackled in the present work is that of meta-learning, that is, given a variable-sized (in $N$) context set $\mathcal{C} = \{(x_i, y_i)\}_{i=1}^N$ of observations from an unknown function $f$, we seek a predictive distribution for outputs at target, or query, locations $x$,

$$p(y \mid x, \mathcal{C}). \tag{2}$$

A strong model for this task should satisfy several desiderata: (i) permutation-invariance to the ordering of context points, (ii) the ability to handle variable context set sizes, and (iii) calibrated uncertainty quantification.

## 2.1 Neural Processes and their limitations

Neural Processes (NPs) and their attention-augmented variants provide a scalable, neural approximation to stochastic processes by learning a mapping from context sets to conditional predictive distributions (Garnelo et al., 2018a; Kim et al., 2019; Gordon et al., 2020). In the canonical NP, predictive uncertainty is read out through a Gaussian output likelihood whose variance is produced by the decoder (Garnelo et al., 2018a).

A single output-variance term captures total predictive uncertainty but does not separate epistemic from aleatoric contributions (Kendall & Gal, 2017), and combined with the documented tendency of early NPs to underfit the context set (Kim et al., 2019), this can yield poorly calibrated predictives. Latent-variable extensions introduce a global stochastic latent to capture function-level variability.

While NPs are attractive for their scalability and ease of amortised inference, early implementations of NPs suffered from practical limitations. Kim et al. (2019) demonstrated that they tended to underfit to signal, and proposed an attention-based alternative, Attentive Neural processes (ANPs), which mitigate this by enabling each query to selectively attend to relevant context points. Transformer Neural Processes (TNPs) (Nguyen & Grover, 2023) further improve the idea by arranging context and target tokens into a single sequence and applying transformer-style blocks. Secondly, although the original NP includes a context-dependent latent variable intended to capture function-level variability, predictive uncertainty is still largely expressed through an output Gaussian variance term, which tends to conflate epistemic and aleatoric uncertainty and can lead to miscalibrated predictions. Again, the TNP improved upon the prior formulations by introducing two methods to impose output space correlations: an autoregressive process and a low-rank non-diagonal covariance matrix. Nguyen & Grover (2023) showed that the autoregressive approach (named TNP-A) has a strong ability to capture functional variability, but lacked smoothness.

## 2.2 Deep Operator Networks

Neural operators learn mappings between function spaces. The Deep Operator Network (DeepONet) (Lu et al., 2021) realises this through two subnetworks whose outputs are combined by an inner product: a branch network, which encodes the input function into a vector of coefficients, and a trunk network, which maps a query coordinate to a vector of basis functions evaluated at that coordinate. For an input function $u$ and a query location $x \in \Omega$, the operator is approximated as

$$\mathcal{G}(u)(x) = \sum_{k=1}^{p} b_k(u)\, t_k(x) + b_0, \tag{3}$$

where $\mathbf{b}(u) = (b_1(u), \ldots, b_p(u)) \in \mathbb{R}^p$ are the branch coefficients, $\mathbf{t}(x) = (t_1(x), \ldots, t_p(x)) \in \mathbb{R}^p$ are the trunk basis functions evaluated at $x$, and $b_0$ is a scalar bias. The trunk outputs can be read as a learned, coordinate-dependent basis $\{t_k(\cdot)\}_{k=1}^{p}$ spanning the output function space, and the branch outputs as the input-dependent coefficients that select a particular function from that function space. The prediction $\mathcal{G}(u)(x)$ is their linear combination. Because the trunk is evaluated pointwise in $x$, predictions are continuous and mesh- or discretisation-free, and hence the operator can be queried at any location without retraining or interpolation. Both networks are trained jointly, typically by minimising a regression loss between $\mathcal{G}(u)(x)$ and target values on paired $(u, x, y)$ examples, where $y$ is an observed target function.

This factored form biases the operator toward a shared, finite-dimensional function space. The trunk learns a common basis over the coordinate domain and the branch need only select coefficients within it, so outputs are assumed to share low-dimensional structure and predictions inherit the smoothness of the basis rather than learning regularity from data. A feature of Equation 3 that we rely on later is the separation of roles, where the branch depends only on the input function $u$, and the trunk only on the query coordinate $x$, with the two interacting solely through the final inner product. In the standard formulation the input function $u$ is assumed to be observed at a fixed set of sensor locations, so that the branch consumes a fixed-dimensional discretisation of $u$. This assumption is what set-based operator variants (Section 3.1) relax, and the branch/trunk separation is what allows one to make only the branch stochastic while keeping the trunk deterministic.

Our contribution is a specific and, to our knowledge, previously unexplored combination. Rather than placing a distribution over operator weights, as in Bayesian DeepONet variants (Garg & Chakraborty, 2022; Lin et al., 2022), or predicting uncertainty from a global latent through a Gaussian output head, as in Neural Processes (Garnelo et al., 2018b; Kim et al., 2019), VSON confines all stochasticity to a low-dimensional, set-encoded branch latent while holding the trunk deterministic. This latent is trained by a proper scoring rule rather than a likelihood, which frees the predictive from a closed-form density assumption and admits non-Gaussian, multimodal behaviour via a normalising flow (Kingma et al., 2017), while the deterministic trunk

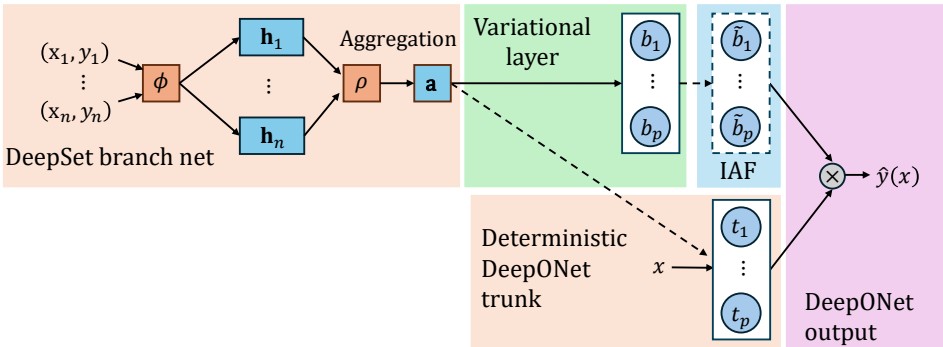

Figure 1: Architecture of the proposed method. The dashed arrow indicates an optional conditioning of the trunk on the contexts' aggregated representation, while the dashed IAF box indicates an optional inverse autoregressive flow layer. If the IAF transformation is not used, $\tilde{\mathbf{b}}$ is the same as $\mathbf{b}$, i.e., the IAF transformation is replaced by the identity function.

preserves smoothness and continuity of function samples. The resulting operator is resolution-independent, retains linear querying complexity (see Section 4.5), and matches or exceeds NP and TNP baselines on calibration while using fewer parameters.

## 3 Methods

The problem is approached from the viewpoint of operator learning, that is, learning a mapping between function spaces. Figure 1 describes the architecture and highlights the various blocks that build upon previous architectures, namely SetONets and DeepONets. Importantly, the SetONet is extended so its branch outputs consist of variational latent variables, which are trained to capture functional uncertainty via their joint distribution.

### 3.1 Set Operator Networks

The standard DeepONet branch assumes the input function is observed at a fixed set of sensor locations, and so cannot accommodate a variable-sized, unordered context set. SetONet (Tretiakov et al., 2025) removes this restriction by replacing the branch with a permutation-invariant set encoder. Each context pair $(x_i, y_i)$ is embedded by a shared MLP $\phi$, and the resulting embeddings are aggregated into a single fixed-size summary,

$$\mathbf{a} = \rho\big(\{\phi(x_i, y_i)\}_{i=1}^N\big) = \rho\big(\{\mathbf{h}_i\}_{i=1}^N\big), \tag{4}$$

where $\rho$ is a permutation-invariant aggregation (Zaheer et al., 2018). A final MLP maps $\mathbf{a}$ to the branch coefficients $\mathbf{b}$, which combine with the deterministic trunk exactly as in Equation 3. SetONets therefore inherit the branch and trunk structure and its inductive bias, while allowing the operator to be conditioned on arbitrary, variable-sized sets of observations.

### 3.2 Variational SetONet

In this work the set-based neural operator described previously is made uncertainty-aware by assuming the branch outputs to be variational variables. Instead of predicting a vector of basis coefficients like in the original SetONet (which itself borrows from the DeepONet architecture), the final branch MLP in the present work predicts distributional terms for each output. Adopting an amortised variational approach (Kingma & Welling, 2014) approach, the branch output vector $\mathbf{b}$ is modelled as a diagonal Gaussian, $q(\mathbf{b}|\mathcal{C}) \sim \mathcal{N}(\mu, \Sigma)$, where the means and diagonal variance terms are outputted by the final branch MLP. We place a standard normal regularising term on the branch latents, $p(\mathbf{b}) \sim \mathcal{N}(0, I)$, which enters the training objective (Equation 10) as a Kullback-Leibler (KL) regulariser, $\text{KL}(q(\mathbf{b}|\mathcal{C})||p(\mathbf{b}))$, to keep the learned distribution $q(\mathbf{b}|\mathcal{C})$ from collapsing to a single value (Titsias, 2009). More details on the training procedure and interpretation of

the objective may be found in Section 3.4. For uncertainty estimation, multiple samples $(\mathbf{b}^{(i)})_{i=1}^{K}$ may be drawn from the variational posterior, combined with the deterministic trunk outputs $\mathbf{t}$ and sample statistics gathered. Note that when the branch terms are independent Gaussian, the output of the inner product becomes a sum of scaled Gaussians, which is itself Gaussian. When modelling non-Gaussian behaviour, the latent variables can be augmented via normalising flows to increase flexibility ($\tilde{\mathbf{b}}$ in Figure 1). More detail can be found in Section 3.3.

**Set Transformer aggregation**   There are a few options for aggregation method $\rho$ in equation 4. SetONet (Tretiakov et al., 2025) uses mean-pooling with self-attention; here we extend it to the Induced Self-Attention Block (ISAB) Set Transformer module (Lee et al., 2019), which allows richer interactions between context embeddings in place of mean pooling.

### 3.3   Further extensions

**Conditioned basis functions**   While the original DeepONet implementation uses independent branch and trunk networks, we allow a conditioning of the trunk network on the aggregated context representations, represented in Figure 1 by the dashed line, to account for the increase in complexity of the problem. A similar structure was adopted by Wang et al. (2021), where they conditioned both networks on the other via cross-connections between MLP layers. In the present work, we chose to condition only the trunk on the contexts to preserve continuous predictive samples. If the branch outputs were conditioned on the coordinate input, each coordinate would have a different latent variational representation, breaking the continuity in output space when making posterior draws.

Conditioning is implemented via layer-wise, context-conditioned modulation of trunk activations using Feature-wise Linear Modulation (FiLM) (Perez et al., 2017). For trunk layer $\ell$ with a post-activation value of $u^{(\ell)}$ the conditioned output is

$$g^{(\ell)}(x) = \gamma^{(\ell)}(\mathbf{a}) \odot \mathbf{u}^{(\ell)}(x) + \beta^{(\ell)}(\mathbf{a}), \tag{5}$$

where $\gamma^{(\ell)}(\mathbf{a}), \beta^{(\ell)}(\mathbf{a}) \in \mathbb{R}^{d_\ell}$ are learned transformation layers dependent on the aggregated context summary $\mathbf{a}$ and $\odot$ denotes the Hadamard product. The final trunk bases $\mathbf{t}(x; \mathbf{a}) \in \mathbb{R}^p$ are then combined with a sampled branch latent $\mathbf{b} \sim q_\omega(\mathbf{b} \mid \mathcal{C})$ via an inner product to produce a scalar value:

$$\hat{y}(x) = \frac{1}{p}\, \mathbf{b}^\top \mathbf{t}(x; \mathbf{a}) + b_0, \tag{6}$$

where the $1/p$ factor stabilises inner-product variance (Lu et al., 2022) and $b_0$ is a learned global bias.

**Inverse autoregressive flows**   To increase the expressivity of the amortised branch over function-level latents we may incorporate inverse autoregressive flows (IAFs) (Kingma et al., 2017). Let the encoder produce a reparameterisable base density

$$q_\omega(\mathbf{b} \mid \mathcal{C}) = \mathcal{N}\big(\mathbf{b};\, \mu_\omega(\mathcal{C}),\, \mathrm{diag}(\sigma_\omega^2(\mathcal{C}))\big), \tag{7}$$

and let $f_\psi : \mathbb{R}^p \to \mathbb{R}^p$ be a composition of autoregressive bijections. The final branch latent is $\tilde{\mathbf{b}} = f_\psi(\mathbf{b})$, whose density follows by change of variables:

$$q_\psi(\tilde{\mathbf{b}} \mid \mathcal{C}) = q_\omega\big(\mathbf{b} \mid \mathcal{C}\big) \left| \det \frac{\partial f_\psi^{-1}(\tilde{\mathbf{b}})}{\partial \tilde{\mathbf{b}}} \right|. \tag{8}$$

IAFs provide a flexible family of variational posteriors that can represent correlations, skewness and multimodality while remaining easy to sample from. Since the flow is invertible, the KL term used for latent regularisation remains computable, so the KL regulariser can still be applied without approximations. There is an important practical trade-off to note. The neural operator maps the latent variables to scalar outputs by a linear combination, that is, the inner product with the trunk terms. If $\mathbf{b}$ is Gaussian then $p(y \mid x, \mathcal{C})$ is Gaussian and many predictive functionals (for example, likelihood) admit closed forms. After a nonlinear flow $\tilde{\mathbf{b}} = f_\psi(\mathbf{b})$ this closed form is generally lost. Predictive evaluation and proper scoring must therefore be estimated by Monte-Carlo sampling from the normalising flow posterior.

Table 1: Results of the 1D regression problem evaluated on three different kernels. We report mean and standard deviation of RMSE on the evaluation sets. RMSE for the VSON models is aggregated per batch, matching the protocol used for the baselines. The conditioned VSON variants achieve the lowest RMSE on the RBF and Matérn kernels, while all VSON variants outperform the baselines on the periodic kernel.

| Method | RBF | Matérn 5/2 | Periodic |
|---|---|---|---|
| ANP | $0.193 \pm 0.001$ | $0.230 \pm 0.002$ | $0.703 \pm 0.002$ |
| BANP | $0.192 \pm 0.001$ | $0.228 \pm 0.001$ | $0.701 \pm 0.008$ |
| TNP-D | $0.177 \pm 0.001$ | $0.222 \pm 0.001$ | $0.664 \pm 0.014$ |
| TNP-A | $0.178 \pm 0.001$ | $0.222 \pm 0.001$ | $0.660 \pm 0.002$ |
| TNP-ND | $0.180 \pm 0.001$ | $0.223 \pm 0.001$ | $0.670 \pm 0.009$ |
| VSON | $0.186 \pm 0.001$ | $0.227 \pm 0.002$ | $\mathbf{0.610 \pm 0.003}$ |
| VSON+IAF | $0.189 \pm 0.003$ | $0.227 \pm 0.001$ | $0.612 \pm 0.008$ |
| VSON+Cond | $0.137 \pm 0.005$ | $0.184 \pm 0.003$ | $0.621 \pm 0.003$ |
| VSON+Cond+IAF | $\mathbf{0.133 \pm 0.005}$ | $\mathbf{0.176 \pm 0.006}$ | $0.621 \pm 0.004$ |

### 3.4 Training the model

We train VSON by minimising an expected scoring-rule loss over held-out targets together with a latent regulariser. Each task is split into two disjoint sets of input-output pairs from the same function $f$. The first is a context set $\mathcal{C}$ (Equation 1), which the model conditions on. The second is a target set $\mathcal{T} = \{(x_j, y_j)\}_{j=1}^m$, which it must predict. We call the target inputs $x_j$ the queries. These are the locations at which the model produces a predictive distribution. The corresponding outputs $y_j$ are withheld from the model and enter only through the loss. During training we resample the partition of each function's observations into context and target sets. This exposes the model to varying numbers and configurations of observed points. The branch induces a variational posterior $q_\omega(\mathbf{b} \mid \mathcal{C})$ over branch latents $\mathbf{b}$, and the inner product with the trunk defines a predictive CDF over the output variable $\hat{y}$ at query $x$

$$F_\theta(\hat{y} \mid x, \mathbf{b}), \tag{9}$$

where $\theta$ denotes all model parameters, which consists of parameters in the deterministic trunk, context encoder and variational parameters.

Instead of assuming a Gaussian noise model on outputs, we use the continuous ranked probability score (CRPS) to match the predicted distributions with data.

$$\mathcal{L}(\theta; \mathcal{C}, \mathcal{T}) =$$
$$\mathbb{E}_{\tilde{\mathbf{b}} \sim q_\psi(\tilde{\mathbf{b}} | \mathcal{C})} \left[ \frac{1}{m} \sum_{j=1}^m \mathrm{CRPS}\big(F_\theta(\cdot \mid x_j, \tilde{\mathbf{b}}), y_j\big) \right] \tag{10}$$
$$+ \beta \, \mathrm{KL}\big(q_\psi(\tilde{\mathbf{b}} \mid \mathcal{C}) \,\|\, p(\tilde{\mathbf{b}})\big),$$

where the CRPS is defined by

$$\mathrm{CRPS}(F, y) = \int_{-\infty}^{\infty} \big(F(\hat{y}) - \mathbb{1}\{\hat{y} \geq y\}\big)^2 \, \mathrm{d}\hat{y}. \tag{11}$$

It's important to note that objective in Equation 10 is not an evidence lower bound (ELBO). Substituting the negative CRPS for a log-likelihood term means the minimiser is a generalised Gibbs posterior in the framework outlined by Bissiri et al. (2016) rather than an approximation to a Bayesian posterior over an evidence term. The scoring rule plays the role of the loss connecting the latent to the data, and the KL divergence to the prior is the coherent regulariser that the framework requires. Training the amortised encoder to minimise Equation 10 over a parametric family $q_\psi$ may be considered an instance of generalised variational inference (GVI) (Knoblauch et al., 2019), where the posterior is defined directly as the minimiser

of an expected loss plus divergence objective over the variational family, with a proper scoring rule as the loss and a KL divergence to the prior. In such a setting, $\beta$ is the temperature scaling the loss against the prior regularisation rather than a fixed unit coefficient, and its value is a calibration choice. We report an ablation over $\beta$ in Appendix B, Table 6.

We adopt the CRPS as a strictly proper scoring rule for evaluating and training distributional predictions, as it measures the discrepancy between the full predictive cumulative distribution and the empirical distribution of targets (Matheson & Winkler, 1976). With flow-augmented branch latents the predictive distribution over outputs has no closed form: the latent flow density is tractable, but its pushforward through the inner product with the trunk is not, so the predictive is accessible only through samples. Likelihood-based training would then require estimating the predictive density at each target from those samples, which is biased and high-variance. CRPS avoids this, because as a strictly proper scoring rule it has an unbiased Monte-Carlo estimator computed directly from predictive samples (Equation 12), and is hence well-defined and stable in exactly the regime where the likelihood is not (Gneiting & Raftery, 2007). When the predictive distribution $F_\theta(\hat{y} \mid x, \mathbf{b})$ is approximated through $K$ Monte-Carlo samples $\{\hat{y}^{(k)}\}_{k=1}^{K}$, $\hat{y}^{(k)} \sim F_\theta(\cdot \mid x, b)$, we use an unbiased Monte-Carlo estimator (Gneiting & Raftery, 2007)

$$
\begin{aligned}
\widehat{\mathrm{CRPS}}\big(F_\theta(\hat{y} \mid x, \tilde{\mathbf{b}}), y\big) = \\
\frac{1}{K}\sum_{k=1}^{K}\big|\hat{y}^{(k)} - y\big| - \frac{1}{2K^2}\sum_{k=1}^{K}\sum_{\ell=1}^{K}\big|\hat{y}^{(k)} - \hat{y}^{(\ell)}\big|,
\end{aligned}
\tag{12}
$$

where the first component encourages fitting to the observed data, and the second encourages spread. Optimisation proceeds over all model parameters $\theta$.

## 4 Results

The proposed method was tested against the state-of-the-art in three different settings: 1D regression, image regression and Bayesian optimisation in 1- and 2-D settings, which constitute typical benchmarks for Neural process evaluation. The method was tested against Transformer Neural Processes, Attentive Neural Processes and Bootstrapping Neural Processes, and used the implementation and hyperparameters of the models provided by Nguyen & Grover (2023). Four different VSON models were used to test for impact and interactions of the extensions described in Section 3.3: with or without IAF on the branch latents and with or without conditioning the trunk on the aggregated context representation.

### 4.1 1D regression

In the 1D regression example, the methods are trained on realisations from a Gaussian Process (GP) prior with an RBF kernel using varying lengthscales and amplitudes. The training and evaluation methodology mirrored the train and evaluation methodology laid out by Nguyen & Grover (2023). Specifically, for each GP realisation lengthscales were sampled from $\ell \sim \mathcal{U}(0.1, 0.6)$[1], and signal amplitudes from $\sigma_f \sim \mathcal{U}(0.1, 1)$. A constant jitter term $\sigma$ was added to the diagonal of the covariance matrix equal to 0.02. The number of contexts and targets for each sampled GP were sampled discretely from $N \sim \mathcal{U}(6, 50)$ and $N \sim \mathcal{U}(3, 47)$, respectively. Training was performed in batches of 16 functions for a total of 100,000 batches. A cosine-decayed learning rate going from $5 \times 10^{-4}$ to $5 \times 10^{-5}$ was used in conjunction with Adam (Kingma & Ba, 2014).

The trained models were evaluated on three different test sets obtained by sampling from GP priors with RBF, Matérn 5/2 and periodic kernels, where the lengthscale and amplitude ranges remained the same as during training, and the periodic kernel additionally sampled its period from $\mathcal{U}(0.1, 0.5)$. Each test set consisted of 48000 functions which were evaluated in batches of size 16, and the metrics averaged over the batches. Results with regard to predictive accuracy are reported in Table 1, while additional metrics detailing uncertainty calibration using CRPS, coverage error (CE), predictive log-likelihood and reliability

---

[1]Although authors of previous works using the benchmark have stated $\ell$ is sampled from $\mathcal{U}(0.6, 1)$, inspection of the code used revealed the interval was as stated in the present text.

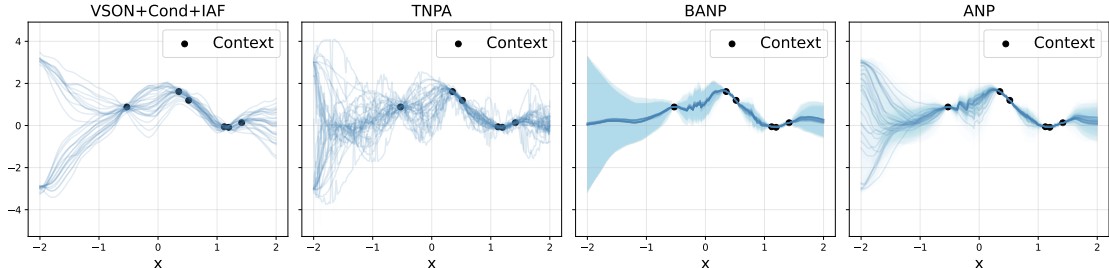

Figure 2: Posterior samples in output space obtained using VSON+Cond+IAF and the comparison models for the conditional GP problem. Filled dots mark the six context points supplied to each model, and the shared anchor at $x = -2$ (value $+3$ or $-3$ with equal probability) causes a bimodal predictive away from the context. VSON recovers both modes, whereas the baselines collapse toward a single mode or the inter-mode mean.

diagrams can be found in Appendices B and D. Conditioning the trunk on the context's aggregation leads to an improvement in predictive accuracy and calibration on the first two kernels, while IAF doesn't have as large an impact. Given that the data is roughly Gaussian for the problem, it follows that augmenting the latents via a normalising flow wouldn't substantially improve results.

Table 2: Results of the 1D multimodal regression problem trained and evaluated on realisations from a conditioned GP. We report RMSE, CRPS and coverage error on the evaluation set. VSON performs best with respect to predictive accuracy.

| Method | RMSE | CRPS | CE |
|---|---|---|---|
| ANP | 0.134 | 0.056 | 0.268 |
| BANP | 0.141 | 0.059 | 0.267 |
| TNP-D | 0.107 | 0.041 | 0.039 |
| TNP-A | 0.115 | 0.042 | 0.047 |
| TNP-ND | 0.111 | **0.0419** | 0.049 |
| VSON+Cond+IAF | **0.096** | 0.048 | **0.035** |

### 4.2   1D multi-modal regression

The model was also trained and tested on a realisations from a GP with conditioned outputs and an RBF kernel. Specifically, a data point was observed at $x = -2$ and with value $y = 3$ or $y = -3$ with even probabilities, and the GP draws were made conditional on the observed value. Outside of the conditioning, training followed the same setup as the previous regression task. Because the two anchor values are equally likely and the model is not shown which was used, the predictive distribution away from the anchor points is a mixture of the two conditioned processes and is therefore bimodal, where one mode tracks the $y = +3$ conditioning and the other the $y = -3$ conditioning. This construction isolates multimodality in a controlled way, so that a model's ability to represent both modes can be assessed directly. To capture the multi-modality in output space and based on the performance in the first regression task, only a VSON model with IAF on the branch latents and a conditioned trunk was used.

Figure 2 represents qualitative performance via samples from the posterior of the trained VSON+Cond+IAF model and 3 comparison models given 6 context points. It highlights the fact that the VSON is able to recover the data-generating process accurately, even with multi-modal outputs. Additional metrics may be found in Table 2, and reliability diagrams are shown in Appendix D.

Table 3: Results of the image regression on EMNIST for seen and unseen classes (digits and letters, respectively). We report mean and one standard deviation for RMSE and CRPS on the evaluation set for 5 initial seeds. The VSON model with both a conditioned trunk and IAF performs best with respect to uncertainty calibration and slightly worse than the state of the art with respect to predictive accuracy on the seen dataset, and best with respect to uncertainty quantification on the unseen dataset. The experiment was run for 5 different initial seeds.

| Setting | Method | RMSE | CRPS |
|---|---|---|---|
| Seen classes (0–9) | ANP | $0.142 \pm 0.000$ | $0.061 \pm 0.003$ |
| | BANP | $0.142 \pm 0.001$ | $0.063 \pm 0.003$ |
| | TNP-D | $0.119 \pm 0.002$ | $0.046 \pm 0.010$ |
| | TNP-A | $0.122 \pm 0.001$ | $0.047 \pm 0.005$ |
| | TNP-ND | $\mathbf{0.116 \pm 0.000}$ | $0.046 \pm 0.002$ |
| | VSON | $0.153 \pm 0.002$ | $0.060 \pm 0.002$ |
| | VSON+IAF | $0.145 \pm 0.001$ | $0.054 \pm 0.002$ |
| | VSON+Cond | $0.119 \pm 0.001$ | $0.039 \pm 0.002$ |
| | VSON+Cond+IAF | $0.118 \pm 0.001$ | $\mathbf{0.037 \pm 0.002}$ |
| Unseen classes (10–46) | ANP | $0.166 \pm 0.001$ | $0.0712 \pm 0.004$ |
| | BANP | $0.161 \pm 0.001$ | $0.0713 \pm 0.003$ |
| | TNP-D | $\mathbf{0.139 \pm 0.001}$ | $0.054 \pm 0.003$ |
| | TNP-A | $0.142 \pm 0.001$ | $0.0555 \pm 0.005$ |
| | TNP-ND | $0.140 \pm 0.001$ | $0.0545 \pm 0.004$ |
| | VSON | $0.185 \pm 0.001$ | $0.076 \pm 0.002$ |
| | VSON+IAF | $0.176 \pm 0.002$ | $0.067 \pm 0.002$ |
| | VSON+Cond | $0.150 \pm 0.001$ | $0.05 \pm 0.002$ |
| | VSON+Cond+IAF | $0.148 \pm 0.001$ | $\mathbf{0.048 \pm 0.002}$ |

### 4.3 Image completion

Image completion was investigated using the EMNIST dataset (Cohen et al., 2017). Each pixel was treated as the value of an underlying image function $y : [-1, 1]^2 \rightarrow [-0.5, 0.5]$ at a coordinate $x \in [-1, 1]^2$. VSON was used as a surrogate model, accepting a context set and a query location as inputs and outputting pixel brightness. The model was trained on the first 10 classes of the dataset, corresponding to the digits 0-9, and evaluated on held-out test sets of the seen classes (digits) and remaining unseen classes (letters). For each task, the number of contexts were sampled from $\mathcal{U}(6, 200)$ and number of targets from $\mathcal{U}(3, 197)$, mimicking the values used in previous studies. Optimisation was carried out using mini-batches containing 100 tasks for a total of 200 epochs. Table 3 displays results on the evaluation sets with respect to predictive accuracy and uncertainty quantification. The results indicate the combination of IAF and conditioned trunk have a large effect on the more complex problem of digit completion, and significantly more so than in the 1D regression problem. Figure 3 presents mean predictions of 3 digits belonging to the observed classes and 3 letters belonging to the unobserved classes. The VSON model with the conditioned trunk produces sharper outputs due to the additional information fed to the basis functions. Comparisons to the state-of-the-art may be found in Section C in the appendix.

### 4.4 Bayesian optimisation

Bayesian optimisation (BO) was carried out in 1-, and 2-dimensional settings. The 1D problem utilised the same setup as the 1D regression problem, namely, VSON was implemented as a surrogate model used to make predictions at new points while accounting for uncertainty. However, the evaluation differed, in that the model was used to optimise black box functions. In the 1D case, the model was trained on GP samples using an RBF kernel, and evaluated on three evaluation sets of synthetic black box functions obtained by sampling from a GP prior using RBF, Matérn 5/2, and periodic kernels. Each evaluation set contained 100 functions, and BO was carried out for 100 iterations. The acquisition function used for the 1D problem was Expected Improvement (EI) (Mockus, 1978) while upper confidence bound (UCB) was used for the 2D

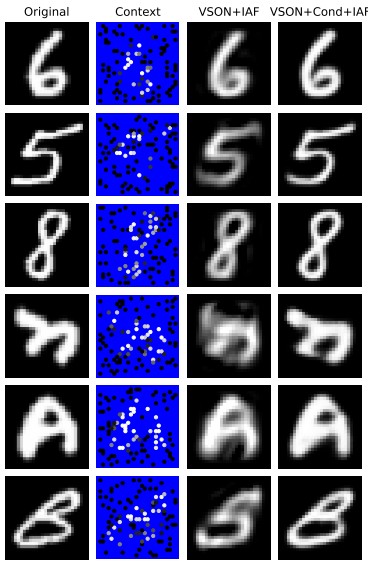

Figure 3: Image completion using 100 context points on the EMNIST dataset. The 2 rightmost columns display mean predictions using VSON+IAF and VSON+Cond+IAF. The first 3 rows display tasks from observed classes, while the latter 3 rows are tasks from unobserved classes. The VSON model with the conditioned trunk produces sharper outputs due to the more informed basis functions.

setting, following the procedure of previous authors, and performance was measured using simple regret. The conditioned trunk VSON model was used for the task, while IAF wasn't, given that the training data came from a GP, which also allowed use of the closed form of EI.

Figure 4a displays results of the 1D BO problem, where VSON performed best across all kernels, mirroring the results of the 1D regression problem. For the 2D problem, the VSON was trained on realisations from a multivariate GP with an RBF kernel, and evaluated on examples borrowed from Bayesian optimisation benchmarks (Kim, 2023). Figure 4b displays the results over 100 different seeds for 100 BO iterations, indicating that VSON was competitive on 2/3 of the problems.

### 4.5 Computational complexity

An attractive property of the proposed method is the separation of the query from the context attention mechanisms present in the branch. TNP-D encounters $\mathcal{O}((N+m)^2)$ complexity at prediction time, where $N$ is the number of context points and $m$ the number of query points, due to the cross-attention mechanisms in the architecture. Although research has been carried out in finding a balance between the flexibility of TNPs and overall complexity (Feng et al., 2023), state-of-the-art neural processes still rely on attention mechanisms between contexts and targets to avoid the under-fitting issues present in the original conditional NP implementation (Garnelo et al., 2018a), which has $\mathcal{O}(m)$ complexity at prediction time. Due to the independence between branch and trunk, VSON shares a $\mathcal{O}(n^2)$ encoding complexity, while retaining an $\mathcal{O}(m)$ querying complexity that matches the cheapest NP baselines, without sacrificing the calibration improvements reported in Section 3. Note that both the base VSON model and the conditioned trunk model share this complexity, as the conditioning vectors described in Section 3.3 depend on the number of contexts only, and then remain fixed for any query.

## 5  Discussion

This work introduces a Variational Set Operator Network, an uncertainty-aware neural operator, to tackle meta-learning problems. The framework combines the meta-learning capabilities inherent in Neural Processes (NPs) with the expressive, continuous function-mapping strengths of Deep Operator Networks. Building

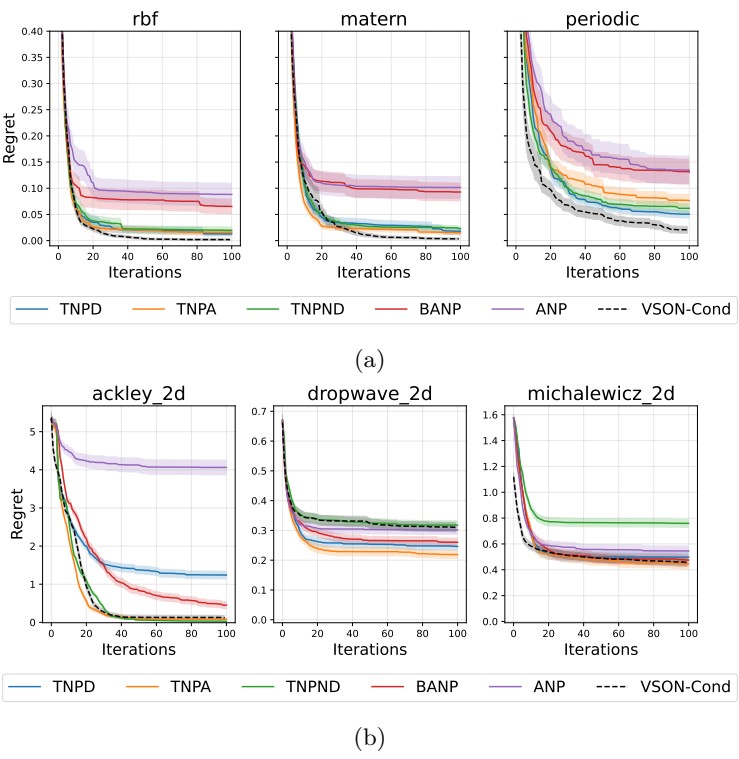

Figure 4: (a) Results of the 1D Bayesian optimisation problem using EI on 3 different evaluation sets. VSON-Cond retrieves the ground truth quicker than the alternatives, and performs well on kernels it wasn't trained on. (b) Results of the 2D Bayesian optimisation problem using UCB on 3 different benchmark problems. VSON-Cond is competitive with state-of-the-art on 2/3 of the benchmarks.

upon previous research which tackles the limitations of standard operators that typically rely on fixed input discretisations, we make the natural extension to account for the uncertainty that follows sparse context. Empirically, VSON improves uncertainty calibration across most settings while remaining competitive on predictive accuracy. It either closely matches or exceeds the Neural Process baselines on 1D regression, multimodal regression, and Bayesian optimisation, and on image completion accepts a small reduction in accuracy in exchange for better calibration.

A core architectural advantage of VSON is the preservation of smoothness and continuity in predictive samples. Unlike autoregressive variants like TNP-A, which may produce jagged predictions due to their sequential nature, VSON utilises a deterministic trunk network that ensures function samples remain smooth with respect to input coordinates. Furthermore, the model offers computational benefits. While attention-heavy models like TNP-A scale quadratically ($O((n + m)^2)$) with the number of context and query points, VSON maintains a linear $O(m)$ querying complexity, albeit a $O(n^2)$ context encoding complexity. Nevertheless, the querying efficiency makes the framework particularly suitable for high-resolution tasks where dense querying is required, for example BO or adaptive design problems in higher dimensions.

Despite these advantages, the transition to non-Gaussian latent spaces introduces a necessary trade-off. The loss of a closed-form predictive distribution requires reliance on Monte-Carlo sampling for inference. Looking forward, the versatility of the VSON framework suggests several promising avenues for extension. Applying the model to physical systems and partial differential equations (PDEs), the original application of DeepONets, where uncertainty quantification is critical, would be a natural progression. A first look at tackling such a problem is given in Section E in the appendix. Exploring incorporating prior knowledge in the model via latent representations such as in the Amortised Conditioning Engine introduced by Chang et al. (2025) may also be an interesting extension, enabling stronger inference.

**Software and Data**

Code is to be made available in the near future.

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

## A  Implementation

Here we report the hyperparameters used for each of the problems. Model dimension $d$ refers to the dimension of the aggregation vector, and latent dimension $p$ serves as the dimension of the inner product output as well. For all problems, the number of samples $K$ to construct the predictive distribution during training was 50, and during evaluation 200, and the scaling parameter $\beta$ of the regulariser was kept at $1 \times 10^{-4}$. When utilising IAF, one IAF layer containing an MLP of depth 2 and using the model dimension width was used. All GPU tasks were run on a single NVIDIA RTX A6000.

**1D regression**

- Model dimension $d$: 64

- Latent dimension $p$: 64

- Number of transformer layers: 2

- Number of transformer heads: 4

- Branch MLP width: 64

- Branch MLP depth: 1

- Trunk MLP width: 64

- Trunk MLP depth: 2

- Activation: ReLU

**1D multimodal regression**  See 1D regression

**Image completion**

- Model dimension $d$: 256

- Latent dimension $p$: 256

- Number of transformer layers: 2

- Number of transformer heads: 8

- Branch MLP width: 256

- Branch MLP depth: 2

- Trunk MLP width: 256

- Trunk MLP depth: 4

- Activation: ReLU

**1D BO**  See 1D regression

Table 4: Parameter counts in the 1D regression problem for the various models.

| Method | Number of parameters |
|---|---|
| ANP | 348418 |
| BANP | 364674 |
| TNP-D | 222082 |
| TNP-A | 220082 |
| TNP-ND | 332821 |
| VSON | 110913 |
| VSON+IAF | 193601 |
| VSON+Cond | 127553 |
| VSON+Cond+IAF | 210241 |

**2D BO**

- Model dimension $d$: 256

- Latent dimension $p$: 256

- Number of transformer layers: 2

- Number of transformer heads: 8

- Branch MLP width: 246

- Branch MLP depth: 2

- Trunk MLP width: 256

- Trunk MLP depth: 4

- Activation: ReLU

Table 4 displays parameter counts for the compared models for the 1D regression problem.

## B   1D regression additional metrics

### B.1   GP prior metrics

Table 5 displays metrics of the 1D regression experiment where 3 different kernels were used to sample from a GP prior. Using IAF in the VSON model didn't appear to have a large impact on the results, while conditioning the trunk lead to slightly improved performance on the first two kernels. The fact that the conditioned trunk model performed worse on the (out-of-distribution) periodic kernel may indicate overfitting, and that the basis functions learnt by the independent trunk tend to be more generalisable.

We additionally report held-out predictive log-likelihood in Table 5. This metric is available in closed form only for the Gaussian-latent variants, since once an IAF is applied the predictive has no closed-form density (see Section 3.3), so likelihood evaluation would require a biased Monte-Carlo density estimate, and we omit the IAF rows rather than report a non-comparable quantity. Despite being trained with CRPS rather than a likelihood, VSON+Cond is competitive on the in-distribution RBF and Matérn kernels, scoring just under TNP-D on RBF (1.39 vs. 1.30). On the out-of-distribution periodic kernel both VSON variants improve substantially over every baseline ($-0.94$ and $-1.86$ against a best baseline of $-2.26$), mirroring the calibration results, where the baselines' likelihoods degrade sharply off-distribution while VSON's remain stable.

Table 5: Results of the 1D regression problem evaluated on three different kernels. We report mean and one standard deviation for RMSE, coverage error (CE), CRPS and predictive log-likelihood on each evaluation set.

| Metric | Method | RBF | Matérn 5/2 | Periodic |
|---|---|---|---|---|
| RMSE ↓ | ANP | $0.193 \pm 0.00$ | $0.230 \pm 0.00$ | $0.703 \pm 0.002$ |
| | BANP | $0.192 \pm 0.00$ | $0.228 \pm 0.00$ | $0.701 \pm 0.008$ |
| | TNP-D | $0.177 \pm 0.00$ | $0.222 \pm 0.00$ | $0.664 \pm 0.014$ |
| | TNP-A | $0.178 \pm 0.00$ | $0.222 \pm 0.00$ | $0.660 \pm 0.002$ |
| | TNP-ND | $0.180 \pm 0.00$ | $0.223 \pm 0.00$ | $0.670 \pm 0.009$ |
| | VSON | $0.186 \pm 0.001$ | $0.227 \pm 0.002$ | $\mathbf{0.610 \pm 0.003}$ |
| | VSON+IAF | $0.189 \pm 0.003$ | $0.227 \pm 0.001$ | $0.612 \pm 0.008$ |
| | VSON+Cond | $0.137 \pm 0.005$ | $0.184 \pm 0.003$ | $0.621 \pm 0.003$ |
| | VSON+Cond+IAF | $\mathbf{0.133 \pm 0.005}$ | $\mathbf{0.176 \pm 0.006}$ | $0.621 \pm 0.004$ |
| CE ↓ | ANP | $0.235 \pm 0.002$ | $0.17 \pm 0.003$ | $0.265 \pm 0.002$ |
| | BANP | $0.236 \pm 0.002$ | $0.17 \pm 0.004$ | $0.217 \pm 0.005$ |
| | TNP-D | $0.043 \pm 0.001$ | $0.045 \pm 0.004$ | $0.129 \pm 0.012$ |
| | TNP-A | $0.045 \pm 0.001$ | $\mathbf{0.044 \pm 0.002}$ | $0.119 \pm 0.008$ |
| | TNP-ND | $0.048 \pm 0.002$ | $0.05 \pm 0.007$ | $0.155 \pm 0.009$ |
| | VSON | $0.045 \pm 0.004$ | $0.048 \pm 0.003$ | $\mathbf{0.088 \pm 0.005}$ |
| | VSON+IAF | $0.044 \pm 0.001$ | $0.057 \pm 0.007$ | $0.100 \pm 0.018$ |
| | VSON+Cond | $\mathbf{0.038 \pm 0.001}$ | $0.066 \pm 0.009$ | $0.134 \pm 0.023$ |
| | VSON+Cond+IAF | $0.039 \pm 0.003$ | $0.064 \pm 0.002$ | $0.137 \pm 0.013$ |
| CRPS ↓ | ANP | $0.084 \pm 0.00$ | $0.109 \pm 0.00$ | $0.463 \pm 0.01$ |
| | BANP | $0.097 \pm 0.00$ | $0.121 \pm 0.00$ | $0.463 \pm 0.01$ |
| | TNP-D | $0.070 \pm 0.00$ | $0.092 \pm 0.00$ | $0.351 \pm 0.02$ |
| | TNP-A | $0.060 \pm 0.00$ | $0.086 \pm 0.00$ | $0.387 \pm 0.02$ |
| | TNP-ND | $0.071 \pm 0.00$ | $0.094 \pm 0.00$ | $0.361 \pm 0.01$ |
| | VSON | $0.075 \pm 0.001$ | $0.102 \pm 0.001$ | $\mathbf{0.318 \pm 0.001}$ |
| | VSON+IAF | $0.077 \pm 0.001$ | $0.102 \pm 0.001$ | $0.321 \pm 0.006$ |
| | VSON+Cond | $0.054 \pm 0.002$ | $0.077 \pm 0.003$ | $0.329 \pm 0.004$ |
| | VSON+Cond+IAF | $\mathbf{0.053 \pm 0.002}$ | $\mathbf{0.074 \pm 0.002}$ | $0.329 \pm 0.003$ |
| Log-Likelihood ↑ | ANP | $0.81 \pm 0.00$ | $0.63 \pm 0.00$ | $-5.02 \pm 0.21$ |
| | BANP | $0.82 \pm 0.01$ | $0.66 \pm 0.00$ | $-3.09 \pm 0.14$ |
| | TNP-D | $1.39 \pm 0.00$ | $0.95 \pm 0.01$ | $-3.53 \pm 0.37$ |
| | TNP-A | $\mathbf{1.63 \pm 0.00}$ | $\mathbf{1.21 \pm 0.00}$ | $-2.26 \pm 0.17$ |
| | TNP-ND | $1.46 \pm 0.00$ | $1.02 \pm 0.00$ | $-4.13 \pm 0.33$ |
| | VSON | $0.90 \pm 0.01$ | $0.37 \pm 0.01$ | $\mathbf{-0.94 \pm 0.06}$ |
| | VSON+Cond | $1.30 \pm 0.04$ | $0.70 \pm 0.07$ | $-1.86 \pm 0.20$ |

## B.2   Effect of the regularisation weight $\beta$

Table 6 reports an ablation over the GVI temperature $\beta$ on the 1D regression problem using the VSON+Cond model. The trend is consistent across metrics. On the in-distribution RBF and on the Matérn kernel the smallest weights ($\beta$ between 0 and $10^{-4}$) give the best accuracy, CRPS and log-likelihood, whereas $\beta = 1$, which may be considered the unit coefficient implied by a naive VAE-style objective, collapses accuracy (RMSE 0.45 on RBF and negative log-likelihood). On the out-of-distribution periodic kernel the ordering reverses, where larger $\beta$ improves both calibration and accuracy, with $\beta = 1$ giving the lowest coverage error and CRPS. This supports treating $\beta$ as a temperature that trades data fit against prior regularisation (as described in Section 3.4) rather than a coefficient with a single correct value. The default $\beta = 10^{-4}$ used throughout sits at a favourable point for in-distribution performance while remaining stable.

Table 6: Ablation across values of $\beta$ on the 1D regression problem, using the VSON+Cond model, which performed well across the model comparison study in Table 5, and admits a closed-form likelihood. The reported values are mean $\pm$ standard deviation across three seeds.

| Metric | $\beta$ | RBF | Matérn 5/2 | Periodic |
|---|---|---|---|---|
| RMSE ↓ | 0 | $0.135 \pm 0.002$ | $0.176 \pm 0.007$ | $0.622 \pm 0.002$ |
| | $10^{-6}$ | $\mathbf{0.134 \pm 0.003}$ | $\mathbf{0.173 \pm 0.002}$ | $0.630 \pm 0.001$ |
| | $10^{-4}$ | $0.137 \pm 0.005$ | $0.184 \pm 0.003$ | $0.621 \pm 0.003$ |
| | $10^{-2}$ | $0.139 \pm 0.003$ | $0.181 \pm 0.003$ | $0.619 \pm 0.002$ |
| | 1 | $0.450 \pm 0.002$ | $0.458 \pm 0.003$ | $\mathbf{0.571 \pm 0.002}$ |
| CE ↓ | 0 | $\mathbf{0.035 \pm 0.002}$ | $\mathbf{0.055 \pm 0.004}$ | $0.113 \pm 0.006$ |
| | $10^{-6}$ | $0.040 \pm 0.002$ | $0.063 \pm 0.002$ | $0.154 \pm 0.011$ |
| | $10^{-4}$ | $0.038 \pm 0.001$ | $0.066 \pm 0.009$ | $0.134 \pm 0.023$ |
| | $10^{-2}$ | $0.161 \pm 0.003$ | $0.108 \pm 0.003$ | $0.084 \pm 0.010$ |
| | 1 | $0.118 \pm 0.004$ | $0.096 \pm 0.004$ | $\mathbf{0.037 \pm 0.006}$ |
| CRPS ↓ | 0 | $\mathbf{0.051 \pm 0.001}$ | $\mathbf{0.073 \pm 0.003}$ | $0.325 \pm 0.002$ |
| | $10^{-6}$ | $0.052 \pm 0.001$ | $\mathbf{0.073 \pm 0.001}$ | $0.336 \pm 0.001$ |
| | $10^{-4}$ | $0.054 \pm 0.002$ | $0.077 \pm 0.003$ | $0.329 \pm 0.004$ |
| | $10^{-2}$ | $0.062 \pm 0.001$ | $0.082 \pm 0.001$ | $0.319 \pm 0.002$ |
| | 1 | $0.216 \pm 0.001$ | $0.223 \pm 0.002$ | $\mathbf{0.300 \pm 0.002}$ |
| Log-Likelihood ↑ | 0 | $\mathbf{1.40 \pm 0.02}$ | $\mathbf{0.80 \pm 0.06}$ | $-1.63 \pm 0.07$ |
| | $10^{-6}$ | $1.36 \pm 0.01$ | $0.74 \pm 0.04$ | $-2.28 \pm 0.33$ |
| | $10^{-4}$ | $1.30 \pm 0.04$ | $0.70 \pm 0.07$ | $-1.86 \pm 0.20$ |
| | $10^{-2}$ | $0.97 \pm 0.01$ | $0.64 \pm 0.03$ | $-0.98 \pm 0.05$ |
| | 1 | $-0.19 \pm 0.01$ | $-0.27 \pm 0.01$ | $\mathbf{-0.88 \pm 0.08}$ |

## C  Image completion comparison figures

Figure 5 displays the mean predictions on 8 tasks from the seen classes and 8 from the unseen classes in the EMNIST dataset using VSON+IAF, VSON+Cond+IAF, TNP-D and TNP-ND.

## D  Reliability diagrams across regression problems

Figure 6 shows reliability diagrams for the three regression problems, plotting empirical coverage against nominal coverage. A perfectly calibrated model lies on the diagonal. Across the first two settings VSON tracks the diagonal closely, consistent with the coverage-error results in Tables 2 and 5. On EMNIST, the Neural process baselines deviate systematically from the diagonal, explained primarily by the unimodal Gaussian predictives of the baselines being unable to reproduce the coverage of a genuinely non-Gaussian target, whereas the VSON performs better.

## E  Physical systems regression

As mentioned in the discussion, an interesting further direction for application is studying physical systems, including problems defined by PDEs.

**Flow around airfoil**  As a real-world benchmark we tackle a problem concerning steady-state flow around 2D airfoils, using a dataset of 1024 simulated aerodynamic fields originating from a benchmark set introduced by Raonic et al. (2023) for neural operator tasks. This appendix is a qualitative demonstration that VSON transfers to data derived from a physical system rather than a benchmarked comparison against neural operator baselines. We report predictions and uncertainty maps to show the method produces spatially structured, physically plausible uncertainty on a PDE-derived problem. Figure 7 displays predictions on an evaluation set. The uncertainty profiles in the rightmost column provide information towards, for example,

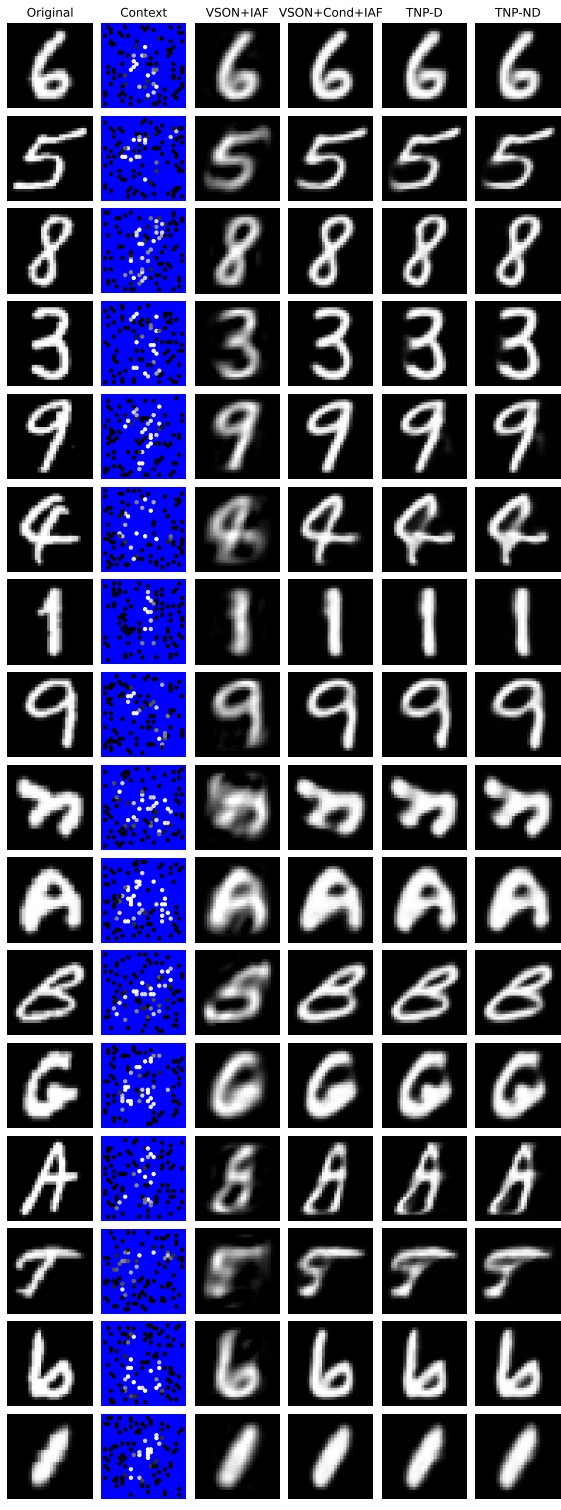

Figure 5: Mean predictions on 8 tasks from the seen classes and 8 from the unseen classes in the EMNIST dataset using VSON+IAF, VSON+Cond+IAF, TNP-D and TNP-ND.

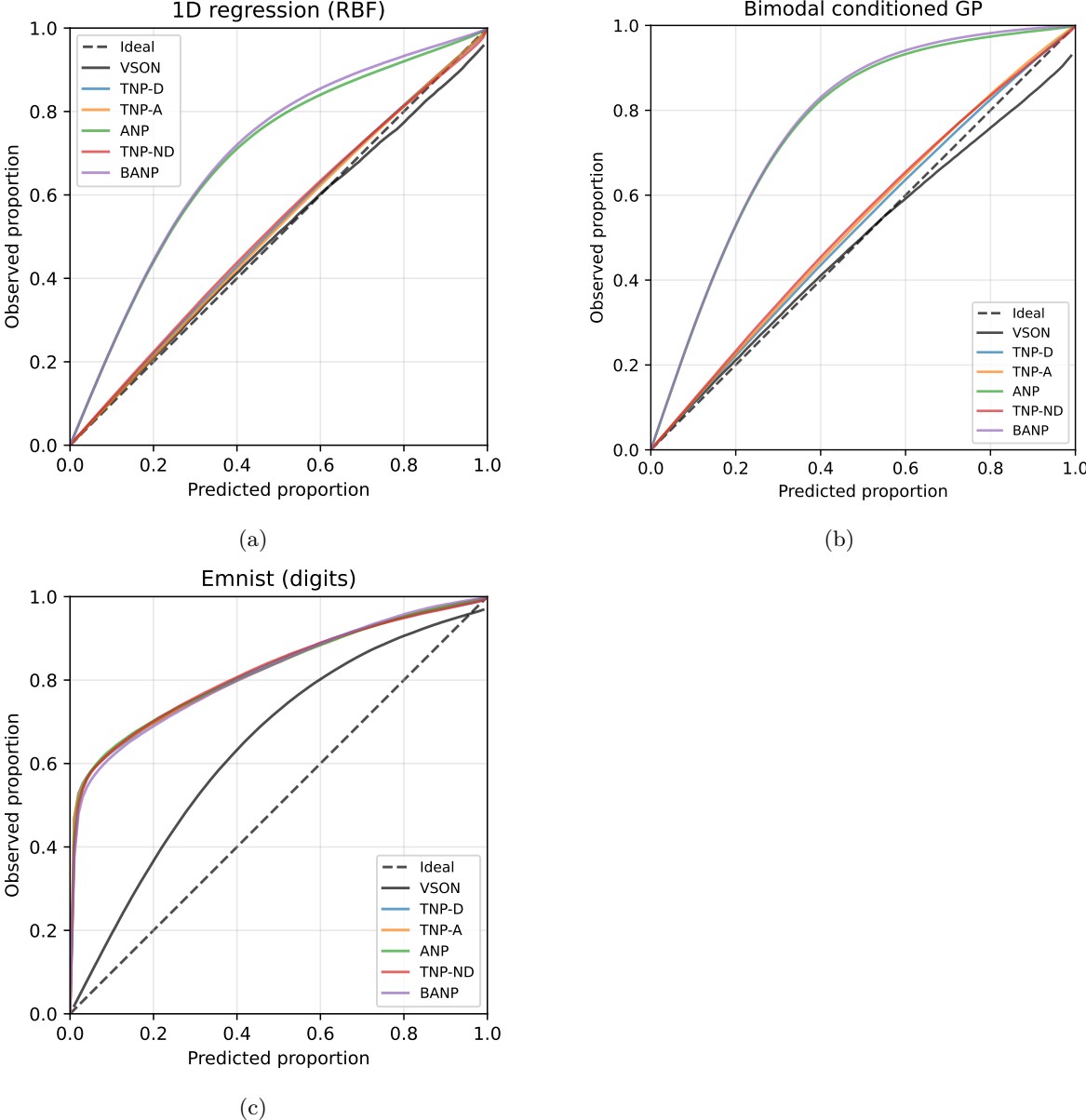

Figure 6: Reliability diagrams for the three regression problems, (a) 1D regression against test set generated using RBF kernel, (b) 1D bimodal regression, and (c) EMNIST digits.

adaptive design of sensor placement. The predictive uncertainty is not spatially uniform. It concentrates in the physically active region around the airfoil and its wake, and it contracts as the context grows from 50 points in the first two rows to 200 in the second two. The uncertainty map indicates where an additional measurement would be most informative, which is the information a sensor placement or adaptive design loop would act on.

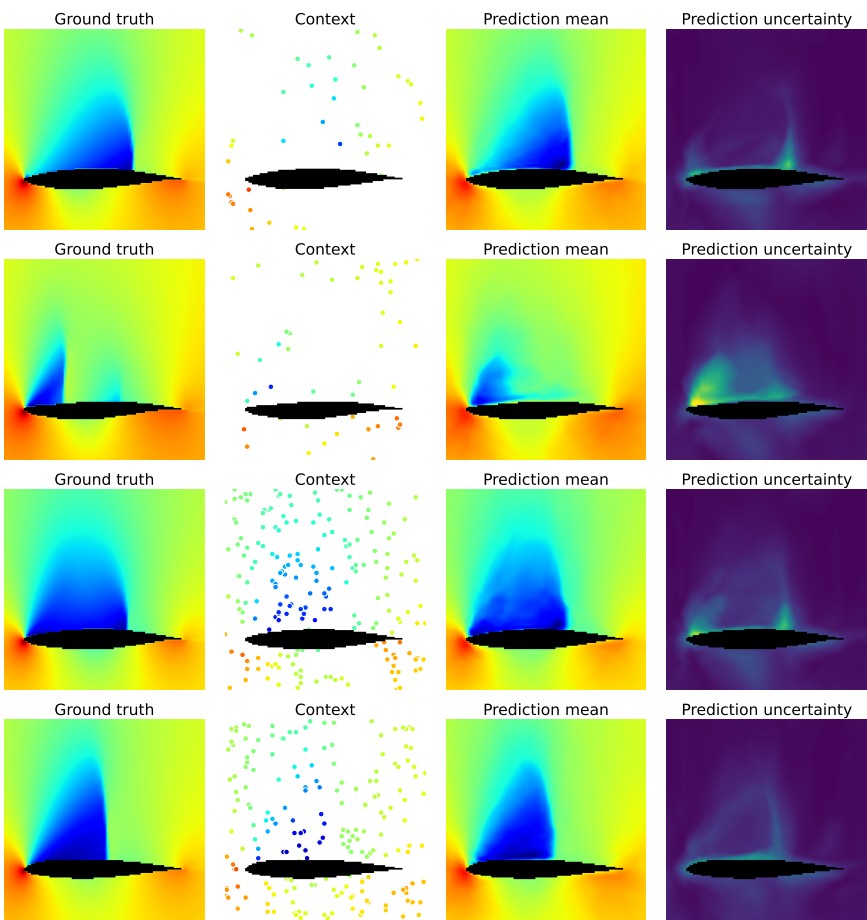

Figure 7: Airfoil problem predictions. The first two rows use 50 contexts, while the second two rows use 200. The uncertainty profiles are on the same scale.

