# OpenReview forum: "Variational Set Operator Networks: Uncertainty-Aware Meta-Learning via Probabilistic Neural Operators"
_TMLR — Under review for TMLR_

### Review · Reviewer_Sdbb · 2026-06-13

**Summary Of Contributions:**

Honestly, I'm kind of surprised I got assigned this paper. I don't know what made the algorithm resort to me. If I had looked harder, I would have flagged to the action editor that I probably shouldn't review this paper. But I didn't, so sorry that you're stuck with me. This is not my area of expertise, so my plan is to understand things as best I can and get through this eval, but then leave a pretty humble confidence score and be ready to listen a little bit more and think a little bit more when the replies come out.

---

I just spent a much longer time attempting without much success to understand what's going on in this paper. Is it my fault? Am I dumb? (probably, and kind of). But it's also just really frustratingly difficult to find a shred of engagement with a practical problem or practical motivation to doing this work. This paper was not meant to be any sort of introduction about operator networks, which I recognize, but it has not succeeded at all in making this stuff very accessible.

My best attempt at a summary of the contributions is to suppose that we have a bunch of data that was generated from some unknown process. We have some sort of class of functions that form our hypothesis class when we are trying to infer what process generated the data. Given i.i.d. data, we want to be able to train neural networks that can sample from that function class and learn to attempt to get to the first inference at what generated the data. This paper starts with a SetONet from prior work and then introduces tricks, including variational inference, an aggregation function, a regularization approach, conditional basis functions, and inverse autoregressive flows. Then, "the proposed method was tested against the state-of-the-art in three different settings: 1D regression, image regression and Bayesian optimisation in 1- and 2-D settings, which constitute typical benchmarks for Neural process evaluation." These seem like pretty simple tasks, and I'm not sure why being able to do this with operator networks is practically useful, but I will take the authors' word that these are reasonable benchmarks. The method in the paepr compares pretty favorably to some baselines.

**Additional Comments:**

It looks like there's not a place to indicate a confidence score, but consider my confidence score 1 out of 5. Sorry.

**Audience:**

No

**Audience Explanation:**

I'm not necessarily proposing any change to the paper, but could the authors explain to me the case for the practical value of this? From my Google scholar search, it seems that function inference given data using neural operators is a relatively popular field. But what's the value in it? Being able to complete the tasks that these benchmarks were created from well doesn't seem particularly impressive. Are there non-operator network baselines for these tasks that are very strong? What about even very simple things like regression on basis functions or naive bayes? If so, should we grade the operator network approach on a curve? When I review deep learning papers, I usually think of it as fair game within that field to question the practical value of the paper and to be concerned with when a paper doesn't accomplish something of much importance. Should I do the same here? Being able to accomplish tasks in this weight class that the paper seems to work in for its benchmarking doesn't clearly seem to me to be very important for the world.

**Broader Impact Concerns:**

None.

**Claims And Evidence:**

Yes

**Claims Explanation:**

I don't notice anything wrong or suspicious about the paper.

**Requested Changes:**

It would have been useful for me in reading the paper if the paper worked on connections and motivations relating to real-world problems. I won't say that I think that the author should go add a bunch of language for people like me and dumb down some of the explanations. But for what it's worth, it might have made my day a little bit easier.

---

> ### Author Response · Authors · 2026-07-09
> **Response to Reviewer Sdbb**
>
> We thank the reviewer for taking the time to engage with the paper despite it falling outside their main area, and for the honesty of the review. The reviewer asked for the practical value of the work to be made clearer, and we agree this was underdeveloped. We address it below and have revised the paper accordingly.
>
> Practical motivation. The setting is amortised inference over a family of functions, where rather than fitting a fresh model per task, a single operator maps a variable-sized set of observations to a calibrated predictive distribution, returning cheap posterior samples at any query location. This matters specifically where observations are expensive or gathered sequentially and where decisions depend on calibrated uncertainty rather than point accuracy alone, for example, Bayesian optimisation of expensive black-box functions, sequential experimental design and active learning, and surrogate modelling of physical systems. We have added a textbf to the introduction stating these applications, and the Bayesian optimisation experiments (Section~4.4) are a direct instance, where there the model is the surrogate, and its calibrated uncertainty is what the acquisition function consumes. We also include a real-world physical systems example in Appendix E, steady-state flow around a 2D airfoil from a published neural operator benchmark, where the predictive uncertainty concentrates in physically meaningful regions and contracts as more observations are supplied.
>
> The baselines and their strength. We would clarify one point that we did not make visible enough. The baselines, ANP, BANP, and the TNP family in the paper, are not operator networks, but rather the dominant neural process and transformer-based meta-learning methods and represent the current state of the art for this task, following the evaluation protocol of Nguyen \& Grover (2023). The comparison is therefore against the strongest available methods for this problem, not weak ones. We would also note (Table 4) that VSON attains its results with fewer
> parameters than the baselines. The base model uses roughly a third of the
> parameters of ANP or BANP and half of TNP-D, and even the largest variant
> (VSON+Cond+IAF) is no larger than any of the compared methods, so the
> performance is not a matter of clearing the bar with a larger model.
>
> Interest to the TMLR audience. We hope the clarified motivation makes concrete what a reader can take from the work, that is, a probabilistic operator that achieves better-calibrated uncertainty than the standard NP and TNP baselines while preserving sample smoothness and linear-in-queries cost. We believe this is of direct interest to the neural process, neural operator, and uncertainty quantification communities, which, as the reviewer's own search indicates, form an active area.
>
> We thank the reviewer again, and are happy to discuss any of the above further.

---

### Review · Reviewer_czEh · 2026-06-19

**Summary Of Contributions:**

The authors develop Variational Set Operator Network (VSON). This operator network involves the following:
 - a latent representation of the context (embedding) through an MLP,
 - a combination of these embeddings in a permutation-invariant manner,
 - a final branch MLP which takes this aggregated embedding and outputs a mean and covariance (i.e. variational distribution over these branch coefficients),
 - an inverse-autoregressive flow that flows this Gaussian distribution into a non-Gaussian one,
 - a deterministic trunk which is a bilinear function of two MLPs (a context-summary dependent function and a query-dependent function),
 - and finally the output is the inner product of this transformed trunk and sampled branch coefficients from the branch posterior.
This paper seems to combine many previous ideas in this field (DeepONets, SetONets, NP, TNP-D, IAF etc.) in a valuable way, and some of the experimental evidence shows good performance of VSON. However, I find it very difficult to understand which aspects of this work are novel, and which components come from previous works. I also believe that a more in-depth presentation of the underlying objects would be beneficial.

**Audience:**

No

**Audience Explanation:**

The experimental results (those without issues detailed above) show improved results, and from what I understand of the work it is a novel addition to the literature, that includes well-thought out changes to the existing frameworks - however, to fully understand the value of this paper, I believe the above issues need to be resolved.

**Broader Impact Concerns:**

No broader impact concerns.

**Claims And Evidence:**

No

**Claims Explanation:**

1. For the most part, I think that VSON is a novel extension of previous works. However, it is hard to understand the explicit combination of previous works. For example, it seems that VB-DeepONet has incorporated variational distributions into this meta-learning framework. Further, the 'original NP includes a context-dependent latent variable intended to capture function-level variability' - how does this differ from your claim that for VSON, 'instead of predicting distributional terms or modelling output space correlations, the uncertainty is learnt via variational variables comprising the branch network outputs' - what are the authors doing differently compared to NP and VB-DeepONet for explicitly the variational component? In another example, do other works use IAF? How common is it to use CRPS? Do other works also condition the trunk functions on the context? As a non-expert in the field, I think a very clear exposition of exactly how you differ from other works would be very enlightening, and show the value of the work.
2. Experimental evidence is mostly good, as it follows the TNP paper. However there are a few issues. Is there a reason the authors choose RMSE instead of NLL (as in the TNP paper) for Table 1? For the experiment in Figure 2, I am confused as to why it is multi-modal, and what the context points are versus the y=3, -3 at x = -2? More explanation would be useful. Finally, could you explain why the results in Figure 6. in the TNP paper for the 2D sets are different to the results in Figure 4. (b) in this work? For example in Michalewicz 2D, this work has TNPND attaining a regret of 0.8, while in the TNP paper it gets 0.3, out-performing the other methods. Also, why did you only include the 2D experiments?

**Requested Changes:**

If a better explanation of how the work expands on / differs from other works in the neural process / operator learning literature is included, and the experimental issues are resolved, then I would be happy to change my recommendation to acceptance.

Some other requests:
 - Many terms are used without definition or citing, for example SetONets, DeepSets.
 - Section 1.1. has subheading 'Neural Processes and amortised learning', but amortised learning is not defined nor discussed in this subsection.
 - The line 'predictive uncertainty is still largely expressed through an output Gaussian variance term, which tends to conflate epistemic and aleatoric uncertainty and can lead to miscalibrated predictions.' is a claim without evidence, reference or explanation.
 - Providing mathematical background on DeepONet (for example the basis and trunk functions, how they are trained, etc.).
 - Line "Unlike likelihood-based training, CRPS remains well-defined and numerically stable when the predictive distribution is only accessible through samples, which is the case for flow-augmented branch latents." should have reference.

---

> ### Author Response · Authors · 2026-07-09
> **Response to Reviewer czEh**
>
> We thank the reviewer for the careful and constructive review. We address each point below.
>
> Distinguishing our contribution from prior work. We agree the borrowed-versus-novel boundary needed to be explicit and have revised the contribution statement accordingly. The organising idea is that VSON confines all stochasticity to a low-dimensional, set-encoded branch latent while holding the trunk deterministic, and trains that latent with a proper scoring rule rather than a likelihood. Section 2.2 now states this directly.
>
> Why the branch alone. The branch selects a function from the learned basis via coefficients produced from the context, while the trunk supplies the basis at a query location. Uncertainty here is uncertainty about which function generated the data, so it belongs on the branch coefficients. With a deterministic trunk, a posterior draw is a single coherent function of the query and samples stay smooth. A stochastic trunk would give each query its own latent draw and discontinuous samples.
>
> VB-DeepONet. The basic difference is the problem solved. VB-DeepONet places a variational distribution over the weights of a single DeepONet, giving weight-space uncertainty for one fixed operator and must be retrained per dataset. VSON is amortised over a family of functions and infers the predictive for a new function from its context set in one forward pass.
>
> The NP latent. A standard NP reads predictive uncertainty out through a Gaussian output-variance head, so its predictive is Gaussian by construction. In VSON the stochasticity lives on the branch coefficients and reaches the output through the inner product with the deterministic trunk. With the flow this predictive is non-Gaussian, enabling the multimodal targets in Section 4.2.
>
> IAF. The flow is standard in flow-augmented VAEs (Kingma et al., 2017) and we adopt rather than introduce it, but it is not standard in neural processes or operators. The contribution is its placement on the branch latent of a set-based operator, turning a Gaussian coefficient posterior non-Gaussian while leaving the trunk and hence sample smoothness untouched.
>
> CRPS. CRPS is well established as a scoring rule. The methodological point is its use as the training objective for an amortised operator whose predictive has no closed-form likelihood, now situated within the generalised variational framework in Section 3.4.
>
> Trunk conditioning. Cross-conditioning has precedent (Wang et al., 2021). We condition only the trunk via FiLM on the aggregated context, for the same reason the trunk stays deterministic. Conditioning the branch on the query would introduce a per-location latent and break sample continuity.
>
> RMSE versus NLL (Table 1). We have added predictive log-likelihood to the 1D regression evaluation (Table 5), following the TNP protocol for direct comparability. LL is exact for the Gaussian-predictive variants, and VSON+Cond is competitive despite training on CRPS.
>
> Figure 2 (multimodality). We have rewritten Section 4.2 and the caption to remove the ambiguity. A single shared anchor at x = -2 takes value +3 or -3 with equal probability, and each GP realisation is drawn conditional on it. Since the model is not shown which value was used, the predictive away from the anchor is bimodal. The six context points come from one such conditioned realisation. Table 2 now also reports CRPS and coverage error across all methods, making the claim quantitative.
>
> The 2D BO results versus Nguyen & Grover (2023). The 2D pipeline runs directly from their released code, so the setup matches theirs, but we could not reproduce the exact TNP-ND regret values of their Figure 6. Training TNP-ND from scratch and running the same BO loop produces markedly different regret curves across runs, indicating the variation is driven substantially by the surrogate's training rather than the BO procedure, so reproducing a specific published curve would require their exact trained checkpoint. We cannot at present fully separate this training variance from any residual configuration difference, and so report the numbers our runs produce rather than tune toward the published values.
>
> Reporting only the 2D setting. The point of the BO study is that VSON's predictive is directly usable within a sequential acquisition loop, where the surrogate's uncertainty is what the acquisition function consumes. The 1D and 2D experiments demonstrate this on standard benchmarks, and extending to 3D follows the same protocol.
> Minor requested changes. We have cited SetONet (Tretiakov et al., 2025) and DeepSets (Zaheer et al., 2018) on first use, defined amortisation in Section 1.1, reworded and referenced the claim about output-variance heads conflating uncertainties (Kendall & Gal, 2017), added DeepONet mathematical preliminaries (Section 2.2), and referenced and sharpened the CRPS stability claim (Gneiting & Raftery, 2007).
>
> We thank the reviewer again. We would welcome confirmation that this matches their expectations.

---

### Review · Reviewer_H57R · 2026-06-29

**Summary Of Contributions:**

This paper addresses meta-learning of conditional function distributions, the problem of producing a predictive posterior distribution given a variable-sized context set of input-output pairs drawn from some unknown function. The paper proposes Variational Set Operator Network (VSON), a probabilistic version of SetONet that replaces deterministic branch coefficients with a variational latent, and an optional inverse autoregressive flow setup to capture non-Gaussian structures. The main claims are that VSON improves uncertainty calibration across benchmarks, that it is competitive in terms of accuracy, and that it captures non-Gaussian behaviour that baselines cannot. While some claims need additional support, the paper is an interesting contribution that will be interesting to TMLR's audience.

**Audience:**

Yes

**Audience Explanation:**

The field of meta-learning seems relevant to TMLR's audience.

**Claims And Evidence:**

No

**Claims Explanation:**

**Strenghts**

- The paper is well-written and clearly motivated. The idea of confining stochasticity to a low-dimensional branch latent while keeping the trunk deterministic is useful and interesting.
- VSON matches or beats baselines with one-half to one-third of their parameters.
- The querying complexity of $\mathcal{O}(m)$ is useful and clearly beneficial.

**Weaknesses**

- I find that the ELBO / variational objective framing is loose. From what I understand, swapping the ELBO's log-likelihood for negative CRPS yields a generalised (Gibbs-posterior) objective, not ELBO. The very small KL weight means the prior is barely enforced. An ablation over $\beta$ would be a useful addition to the paper.
- The evaluation is biased by the choice of metric. VSON is trained on CRPS while the baselines are trained on log-likelihood, yet CRPS is then used as a primary calibration metric. This compares the models on a loss that only one of them optimises. The standard Neural Process metric, predictive log-likelihood, is not reported anywhere. Its absence removes the most natural basis for comparison and makes it impossible to situate the results against published Neural Process numbers.
- The claim about multimodality is not quantified. Table 2 reports only RMSE, which is minimised by the mean lying between the two modes and therefore cannot truly show a model capturing both modes. Multimodality is shown only qualitatively, and only for a single variant. A proper scoring rule or likelihood metric, reported across all baselines would strengthen this claim.

**Requested Changes:**

Can you please provide the requested ablation on $\beta$ and address the weaknesses mentioned above?

---

> ### Author Response · Authors · 2026-07-09
> **Response to Reviewer H57R**
>
> We thank the reviewer for their detailed and technically engaged review, which has materially improved the paper, in particular the characterisation of the training objective.
>
> The objective framing (ELBO / generalised posterior). The reviewer is right that the ELBO description was imprecise. Substituting negative CRPS for a log-likelihood term does not give an ELBO, and the minimiser is in fact a generalised (Gibbs) posterior in the framework of Bissiri et al. (2016), not an approximation to a Bayesian posterior over an evidence term. We have rewritten the relevant part of Section 3.4 to state this, where the scoring rule plays the role of the loss, the KL to the prior is the coherent regulariser the framework requires, and training the amortised encoder over a parametric family is an instance of generalised variational inference (Knoblauch et al., 2019). We have removed the ELBO-objective wording throughout. Under this framing $\beta$ is the temperature weighting the loss against the prior regularisation rather than a fixed unit coefficient, which also motivates the ablation below.
>
> The $\beta$ ablation. We have added this in Appendix B, Table~6. It
> sweeps $\beta$ over a range of values on the 1D regression
> problem, using VSON+Cond so that closed-form likelihood metrics are available,
> and reports RMSE, CE, CRPS and LL on all three kernels. The behaviour is
> consistent with the temperature reading. In-distribution performance on RBF
> and Mat\'ern is stable and near-best for $\beta$ up to $10^{-4}$, degrades at
> $10^{-2}$, and collapses at $\beta = 1$, where the KL dominates the
> scoring-rule loss and the model underfits toward the prior. In-distribution
> RMSE roughly triples at $\beta = 1$ relative to the small-$\beta$ settings.
> Our choice of $10^{-4}$ reflects in-distribution performance rather than a
> claim that it optimises out-of-distribution calibration. Raising the weight
> from $0$ to $10^{-4}$ costs nothing in-distribution, with RMSE, CRPS and LL
> on RBF and Mat\'ern within noise of the $\beta = 0$ values. On the periodic
> kernel small weights do not clearly improve calibration over $\beta = 0$;
> substantial gains appear only at $10^{-2}$ and $1$ (CE $0.084$ and $0.037$),
> at a clear in-distribution cost (RBF CE $0.161$ at $10^{-2}$, RMSE collapse
> at $\beta = 1$). The ablation thus exhibits the trade-off the temperature
> reading predicts, where the prior buys robustness under kernel shift at the price
> of in-distribution fit, and no single value dominates both regimes. This
> also answers the point that a small KL weight barely enforces the prior.
> That is accurate and intended, as $\beta = 10^{-4}$ keeps the scoring-rule
> loss dominant on the training distribution, and the ablation makes the
> cost of enforcing the prior more strongly an explicit, characterised choice.
>
> Predictive log-likelihood. We agree this was a missing
> comparison and have added it. Table~5 now reports predictive LL alongside
> RMSE, CE and CRPS on the 1D regression kernels, following the TNP protocol
> for direct comparability. We report LL for the Gaussian-latent variants
> (VSON, VSON+Cond) only, where the predictive density is exact; the IAF
> variants have no closed-form density, and reporting only exact values
> avoids mixing exact and sampled likelihood estimates across tables.
> VSON+Cond is competitive with the likelihood-trained baselines despite
> training on CRPS, roughly matching TNP-D on RBF, and on the
> out-of-distribution periodic kernel both variants improve substantially
> over every baseline ($-0.94$ and $-1.86$ vs.\ a best baseline of $-2.26$),
> where likelihood-trained models degrade sharply under the kernel shift.
>
> Multimodality claim. The reviewer is correct that RMSE alone
> cannot show a model capturing both modes, and that this was previously only
> qualitative. Table 2 now reports CRPS and coverage error across all methods
> on the multimodal task, in addition to RMSE. VSON attains the lowest
> coverage error ($0.035$, against $0.039$--$0.049$ for the TNP variants and
> $0.267$--$0.268$ for ANP and BANP) and the lowest RMSE ($0.096$), because
> it spreads mass over both modes rather than committing to one, as the
> posterior samples in Figure 2 show directly. The likelihood-trained
> transformers score better on CRPS (TNP-D $0.041$ vs.\ VSON $0.048$), which
> rewards sharp prediction on the realised mode; coverage error is the
> property that reflects whether a model accounts for the mode it was not
> shown, and is where VSON's advantage lies.
>
> We thank the reviewer again and are happy to answer any further questions.

---

### Author Response · Authors · 2026-07-09
**Summary of Revisions**

We thank all reviewers for the detailed and constructive reviews, and for setting out the concerns clearly. The revised manuscript makes the following principal changes. The training
objective in Section 3.4 is now correctly characterised as a generalised
variational objective in the sense of Bissiri et al. (2016) and Knoblauch
et al. (2019) rather than an ELBO, with $\beta$ interpreted as the
temperature weighting the scoring-rule loss against the prior regulariser.
An ablation over $\beta$ has been added (Appendix B, Table 6), and
predictive log-likelihood is now reported alongside the existing metrics
for the closed-form VSON variants (Table 5). Section 2.2 adds mathematical
preliminaries on the DeepONet branch--trunk decomposition, and the
contribution statement now delineates explicitly which components are
inherited from prior work and which are novel. The multimodal regression
claim is now supported quantitatively (Table 2, coverage error and CRPS)
in addition to the qualitative samples in Figure 2. Detailed responses to
each reviewer follow in the individual threads.

Additionally, while preparing this revision we identified an inconsistency in the evaluation code for the first 1D kernel (RBF, Matern, periodic) regression experiment specifically. In that experiment VSON's RMSE was pooled across all evaluation functions, whereas the baseline figures (following Nguyen and Grover, 2023) are averaged per batch of 16. We have corrected VSON's RMSE to the same per-batch protocol for a like-for-like comparison. The corrected values now appear in Tables 1 and 5, and are more in line with the pattern of the following experiments. The best VSON model's RBF RMSE changes from 0.095 to 0.133 and remains the best of all compared methods, so the accuracy claim is unaffected, but is more in line with the performance seen across the other benchmarks. CRPS and predictive log-likelihood are computed per batch and are unchanged. The issue was confined to the 1D regression evaluation code, so the multimodal regression, image completion and Bayesian optimisation experiments use separate evaluation routines and are unaffected.